# ADAPTIVE UNCERTAINTY-AWARE REINFORCEMENT LEARNING FROM HUMAN FEEDBACK

## ABSTRACT

Reinforcement learning from human feedback (RLHF) is a popular technique to align large language models (LLMs) to human preferences. It requires learning a reward model that predicts scalar values given a generated text sequence, acting as a proxy for human preference scores. A central problem of RLHF is *reward hacking*, i.e., overoptimization. LLMs can easily exploit the reward model by generating text that can receive high scores but no longer align with human preferences. We address this problem by proposing a new objective which adapts the tradeoff between reward model score and regularisation based on reward uncertainty. We hypothesize that when the reward model uncertainty is low, RLHF should make a larger step size by lowering the regularization coefficient. On the other hand, when the uncertainty is high, optimization should slow down by staying closer to the original model. We present a novel re-formulation of the RLHF objective and derive our approach from its generalization to account for reward model variance. We demonstrate that our uncertainty-aware RLHF objective mitigates overoptimization and outperforms vanilla RLHF by 50% on a standard summarization task.[1]

## 1 INTRODUCTION

A popular way to align large language models (LLMs) to human preferences is to perform preference optimization via Reinforcement Learning from Human Feedback (RLHF, Ziegler et al., 2020). This enables LLMs to obtain superior performance compared to vanilla fine-tuned models.

RLHF learns a reward model on human-annotated preference data and uses it as a proxy for how humans would score LLM responses. However, a proxy reward model is not a perfect substitute for humans. It typically works well in early optimization iterations when LLM responses are similar to those in its training data. As the LLM responses change during RLHF, the proxy reward model becomes increasingly inaccurate, opening up possibilities for the LLM to exploit the reward model errors. For example, non-sensical responses can potentially be scored highly by the reward model purely by chance. The LLM, erroneously guided by the reward model, overfits to these errors, and the actual quality of its responses starts to decrease. This phenomenon is commonly called *reward hacking* or *overoptimization* (Gao et al., 2023; Eisenstein et al., 2023). We illustrate this with an example in Figure 1.

Ideally, we would like to detect when the reward hacking starts happening and stop the optimization before the LLM's quality decreases. Since we do not have access to the actual reward (i.e., real human preferences), it is common to instead regularize the LLM so it does not shift too far from its original parameters and stop the training once it reaches a certain threshold (Stiennon et al., 2020). However, this regularization penalty is given a fixed weight throughout RLHF and for all samples. Hence, if the proxy reward is high enough at some point during optimization, the reward hacking still occurs (Gao et al., 2023). Vice-versa, this term may also prevent the model from following actually good rewards when these are indeed aligned with human preferences. Additionally, choosing a training stopping point arbitrarily might come too early or too late to reach the full alignment potential.

---

[1]Code will be made publicly available.

We address the above-mentioned issues by incorporating reward model uncertainty into the RLHF objective, naturally resulting in two types of adaptivity:

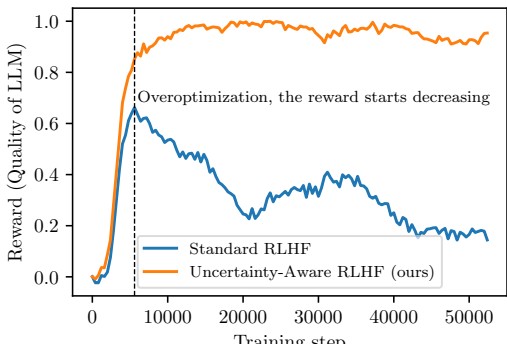

1. The regularization component of the objective is scaled according to the reward model variance, resulting to stronger regularization when the confidence of the reward model is low and, vice-versa, vanishing when the confidence is high. This allows the LLM to optimize for rewards that are expected to be aligned with human preferences and ignore those that are expected to be errors.

2. The whole objective is scaled by the inverse of the reward model's variance, leading it to vanish (and hence the gradient) as the reward model becomes more uncertain. We show that this has an automatic early-stopping effect.

Figure 1: The problem of reward hacking when optimizing for the proxy reward: after a while, the LLM learns to exploit reward model misspecifications, and the actual reward decreases. Ideally, we want the LLM to slow down and early stop the training once this situation occurs, not to regress (our contribution).

These two adaptive features of our method derive directly from our novel theoretical contribution. We first realize that the standard RLHF objective can be interpreted as a product of experts (PoE) of two Gaussians, one predicting reward from the reward model and the other from proximity to the starting model, but both with fixed variance. We argue that this fixed variance assumption is too restrictive, as the relative variance of experts in a PoE is crucial in determining how they combine (Hinton, 1999). We therefore relax this assumption and introduce the variance of the first expert, measured as the reward model variance. This generalization naturally leads to the two adaptive features detailed above. In our experiments, we demonstrate on a summarization task how our adaptive method achieves higher rewards compared to several baselines and also greatly mitigates the reward hacking effect, maintaining high rewards throughout optimization (see Figure 1).

## 2 BACKGROUND

RLHF (Ziegler et al., 2020) consists of three main stages: (1) preference collection; (2) reward model training; and (3) reinforcement learning.

**Collecting preferences.** Initially, two responses are sampled from a supervised fine-tuned LLM for a given prompt. To increase the diversity, the responses can also be sampled from different LLMs. A human annotator is asked to choose the preferred response over the two sampled choices. This step is repeated $n$ times to collect the preference dataset $\mathcal{D} = \{(x_i, y_{i,+}, y_{i,-})\}_{i=1}^{n}$, where $x_i$ is the prompt and $y_{i,+}$ and $y_{i,-}$ are the prefered and rejected answers. In recent work, it is common to have another larger LLM, e.g., GPT-4 (Bubeck et al., 2023), to replace humans in preference annotation to speed up the process and reduce the costs (Dubois et al., 2023).

**Reward model training.** A reward model $\phi(x, y)$ is trained on the preference data $\mathcal{D}$ to assign a score to a given prompt-response pair $(x, y)$. The reward model is usually initialized from a supervised fine-tuned LLM, and its language modeling head is replaced with a linear layer that outputs a single scalar. The model is then trained with the following loss

$$\begin{aligned}
\mathcal{L}_\phi = &-\log \sigma(\phi(x, y_+) - \phi(x, y_-)) \\
&+\eta(\phi(x, y_+) + \phi(x, y_-))^2,
\end{aligned} \tag{1}$$

where $\sigma(x) = (1 + \exp(-x))^{-1}$ is the logistic function. The first term is the difference between the reward inferred for the preferred and rejected answers, which we aim to maximize. The second

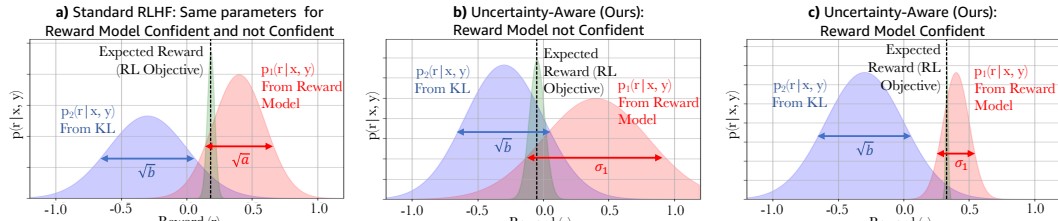

Figure 2: RL objective interpreted as an expectation of reward under a Gaussian product of experts (PoE). The standard RLHF objective **(a)** derives from assuming both experts have fixed variance, leading to the expected reward (green) being unaffected by the reward model confidence. In our uncertainty-aware strategy **(b-c)** we estimate the reward model variance $\sigma_1^2$ and incorporate it. This moves the expected reward towards the expert from the KL divergence (blue) when the reward model is not confident **(b)**, and towards the reward model expert (red) when it is confident **(c)**.

term in the loss is added, so the rewards are centered around zero, where $\eta$ is a small positive value. The reward model is usually trained over a single epoch to prevent overfitting.

**Reinforcement learning.** Finally, we train the LLM, denoted by $\pi_\theta$, to maximize the reward model score. We also keep the original LLM with weights frozen, denoted by $\pi_{\text{ref}}$, and penalize $\pi_\theta$ if its responses diverge from $\pi_{\text{ref}}$. This results in the following optimization objective to be maximized:

$$\arg\max_\theta \mathbb{E}_x\big[\mathbb{E}_{\pi_\theta(y|x)}[\phi(x, y)]$$
$$-\lambda D_{\text{KL}}(\pi_\theta(y \mid x)\|\pi_{\text{ref}}(y \mid x))\big] \qquad (2)$$

where $\mathbb{E}_x$ and $\mathbb{E}_{\pi_\theta(y|x)}$ indicate expectations, and therefore sampling, from the data set of inputs $x$ and the LLM outputs $y \sim \pi_\theta(y \mid x)$ respectively. $D_{\text{KL}}(\cdot\|\cdot)$ denotes the Kullback–Leibler divergence between two distributions (KL-penalty, Kullback & Leibler, 1951) and $\lambda$ is the regularization coefficient. The reason behind keeping $\pi_\theta$ close to $\pi_{\text{ref}}$ through the KL divergence is that $\phi$ is increasingly inaccurate as $\pi_\theta$ diverges from the original distribution where $\phi$ was trained. The optimization is usually done by a policy gradient algorithm, such as proximal policy optimization (PPO, Schulman et al., 2017; Stiennon et al., 2020) or REINFORCE (Williams, 1992; Ahmadian et al., 2024).

## 3 METHOD

Despite using the KL-penalty (Eq. 2), $\pi_\theta$ learns to exploit $\phi$ and finds the responses that are scored highly by $\phi$ but poorly by humans. Therefore, the overall performance starts to decrease. One way to mitigate the reward hacking is to increase the KL-distance regularization coefficient lambda $\lambda$. However, it is difficult to set $\lambda$ optimally. When $\lambda$ is set too high, it leads to inefficient training where $\pi_\theta$ stays too close to $\pi_{\text{ref}}$.

To address this, we propose an adaptive objective function that modifies Eq. 2 to allow for (i) an adaptive KL coefficient, which changes regularization according to reward uncertainty; and (ii) implicit early stopping. These adaptive features derive from our novel re-formulation of standard RLHF as a special case of products of experts (PoE) of two reward distributions and its generalisation to include reward model uncertainty.

### 3.1 RE-FORMULATION OF THE RL OBJECTIVE

We show that the standard RL objective of Eq. 2 is a special case of the expectation of reward $r$ from a product of two Gaussian reward distributions $p_1(r|x, y)$ and $p_2(r|x, y)$. The two distributions, or experts, give two independent estimates of the reward given the input-output pair $(x, y)$ and combine their estimate with an AND operator through a product. This is known as a product of experts (PoE) (Hinton, 1999) and we schematically show the concept in Figure 2. Consider the following general

form for the expected reward from a distribution constructed as a PoE of two arbitrary Gaussians, given an input prompt $x$:

$$\mathbb{E}_{\pi_\theta(y|x)}\mathbb{E}_{p_1(r|x,y)p_2(r|x,y)} r =$$

$$\mathbb{E}_{\pi_\theta(y|x)} \int \mathcal{N}(r; \mu_1, \sigma_1^2)\mathcal{N}(r; \mu_2, \sigma_2^2)r dr = \tag{3}$$

$$\mathbb{E}_{\pi_\theta(y|x)} \frac{\sigma_2^2\mu_1 + \sigma_1^2\mu_2}{\sigma_1^2 + \sigma_2^2}.$$

Here $r$ is the reward, the expectation of which we wish to maximize, and $\mathcal{N}(\cdot; \mu, \sigma^2)$ indicates a univariate Gaussian distribution with mean $\mu$ and variance $\sigma^2$. A detailed derivation is shown in appendix B.1. Now, we set the parameters of the two Gaussians to specific values:

$$\mu_1 = \phi(x, y), \quad \mu_2 = \log \frac{\pi_{\text{ref}}(y \mid x)}{\pi_\theta(y \mid x)},$$

$$\sigma_1^2 = a, \quad \sigma_2^2 = b. \tag{4}$$

Here, $a$ and $b$ are positive constants independent of the prompt $x$ and generated text $y$. With these parameters, the first expert $p_1(r|x, y)$ derives its mean prediction from the reward model $\phi(x, y)$ and assumes constant variance $a$. The mean of the second expert $p_2(r|x, y)$ dictates that the higher the probability under the original weights $\pi_{\text{ref}}$, the higher the reward, which introduces regularisation. Its variance is also constant. Maximizing the expectation of the reward in Eq. 3, we obtain:

$$\arg\max_\theta \mathbb{E}_x[\mathbb{E}_{\pi_\theta(y|x)}\phi(x, y)$$

$$- \frac{a}{b}D_{KL}(\pi_\theta(y \mid x)\|\pi_{\text{ref}}(y \mid x))]. \tag{5}$$

The proof is given in Appendix B.2. The above objective is equivalent to that of Eq. 2, with $\lambda$ set to $a/b$. Therefore, the standard RLHF objective can be interpreted as an expected reward maximization under a PoE of Gaussian reward distributions, of which the variances are both assumed to be constants $\sigma_1^2 = a$ and $\sigma_2^2 = b$.

## 3.2 RELAXING THE FIXED VARIANCE ASSUMPTION

We argue that the assumption of fixed variances in the above formulation is too restrictive and does not take into account the measurable variance of the reward model, which can be used to capture confidence in the reward predictions. In fact, as shown in Figure 2, variance plays an important role in PoEs, essentially adapting the importance of each expert based on their relative confidence (Hinton, 1999). We therefore compute the moments of the first expert $\mu_1$ and $\sigma_1^2$ to be the mean and variance of the reward model output, given an input $x$. To properly capture the uncertainty in the reward model, we employ a deep ensemble of $M$ models $\phi_m(x, y)$, each trained with a different random seed. This approach has been shown to be effective at capturing model uncertainty (Lakshminarayanan et al., 2017) and has been proven to improve robustness in RLHF (Coste et al., 2024). The moments of $p_1(r|x, y)$ are then computed as follows:

$$\mu_1 = \bar{\phi}(x, y) = \frac{1}{M}\sum_m^M \phi_m(x, y),$$

$$\sigma_1^2 = \mathbb{E}_{sg[\pi_\theta(y|x)]}\frac{1}{M}\sum_m^M (\bar{\phi}(x, y) - \phi_m(x, y))^2, \tag{6}$$

where $sg[]$ indicates the stop-gradient operator. We make two approximations in defining $\sigma_1^2$ above; i) this variance of the reward model is approximated by marginalising over generations $y$ and ii) the gradient is stopped from propagating to the generative model $\pi_\theta(y \mid x)$. These approximations allow us to define a practical objective function for our method that can be readily implemented with existing RLHF infrastructue (details in Appendix B.3). With these parameters, and leaving the moments of $p_2(r|y, x)$ the same, we can now derive our objective function from Eq. 3. To simplify notation, we write $D_{\text{KL}}$ instead of $D_{\text{KL}}(\pi_\theta(y \mid x)\|\pi_{\text{ref}}(y \mid x))$ and $\pi_\phi$ instead of $\pi_\theta(y \mid x)$:

$$\arg\max_\theta \mathbb{E}_x \left[ \frac{b}{b + \sigma_1^2} \left( \mathbb{E}_{\pi_\theta}\mu_1 - \frac{\sigma_1^2}{b}D_{\text{KL}} \right) \right] \tag{7}$$

---

**Algorithm 1** Uncertainty-aware adaptive RLHF

---

**Input:** preference dataset $\mathcal{D}$, SFT $\pi_\theta$

{# *Training $M$ reward model ensembles*}

**for** $m \in [M]$ **do**

    Set random seed to $m$ and reshuffle $\mathcal{D}$

    Initialize $\phi_m$ from $\pi_\theta$

    Replace the $\phi_m$ LM head with a linear head

    Train $\phi_i$ with $\mathcal{L}_\phi$ from Eq. 1 on $\mathcal{D}$

**end for**

{# *Reinforcement learning*}

**for** batch $\mathcal{D}_B \in \mathcal{D}$ **do**

    $y_i \sim \pi_\theta(\cdot \mid x_i)$ for $x_i \in \mathcal{D}_B$

    $\mu_{1,i}, \sigma_{1,i}^2 \leftarrow$ Eq. 6

    $b \leftarrow \frac{\mathbb{E}_i[\sigma_i^2]}{\lambda}$ {# Fix this during the first batch}

    $\arg\max_\theta \mathbb{E}_x \left[ \frac{b}{b+\sigma_1^2} \left( \mathbb{E}_{\pi_\theta} \mu_1 - \frac{\sigma_1^2}{b} D_{\text{KL}} \right) \right]$

**end for**

---

The proof is given in appendix B.4. The above objective function naturally introduces two types of adaptivity. Firstly, the KL coefficient $\sigma_1^2/b$ becomes larger with a larger variance in the reward model ensemble $\sigma_1^2$. This results in stronger KL regularization when the reward model is unsure about its estimates. Secondly, the whole objective is scaled by $b/b + \sigma_1^2$, resulting in the objective, and hence the gradient, to be smaller with a larger reward model variance. This introduces an early stopping effect, where the model updates vanish as the reward model becomes more uncertain. These two intuitively desirable adaptive effects are entirely derived from our novel re-formulation and generalization of the RLHF objective. They are relatively straightforward to apply in practice.

Our uncertainty-aware adaptive RLHF method is described in Algorithm 1. First, we train an ensemble of $M$ reward models $\phi_m$ by using a different seed when initializing the linear head and shuffling the data. For reinforcement learning, we sample the first batch of prompts $\mathcal{D}_B$ from $\mathcal{D}$ and generate a response $y_i \sim \pi_\theta(\cdot \mid x_i)$ for each $x_i \in \mathcal{D}_B$. We score each prompt-response pair with all $M$ reward models and calculate its mean $\mu_{1,i}$ and variance $\sigma_{1,i}^2$. To compute the hyperparameter $b$, we use mean variance over the first batch so that the regularization coefficient $\frac{\mathbb{E}_i[\sigma_{1,i}^2]}{b} = \lambda$ during the first batch. We plug $\mu_{1,i}$ and $\sigma_{1,i}^2$ into Eq. 7 and use this objective in RL optimization.

## 4 EXPERIMENTAL SETUP

Figure 3 illustrates the full RLHF optimization pipeline.

**Pretrained models.** We experiment with two GPT-2 (Radford et al., 2019) models (137M and 380M parameters) for reinforcement learning. We also use a larger GPT-J (Wang & Komatsuzaki, 2021) model (6B parameters) as the gold reward model. We limit model size to fit within our computational budget as reinforcement learning is an expensive procedure (Touvron et al., 2023).

**Task & Data.** Following related work (Stiennon et al., 2020; Zhai et al., 2023; Zhang et al., 2024b), we use a filtered version[2] of the Reddit TL;DR summarization dataset (Völske et al., 2017) for Reddit posts summarization. This includes summaries between 20 and 48 tokens and subreddits understandable to the general population. The final dataset consists of 129,722 posts, with 2,000 held out as a validation set.

**Supervised Fine-tuning.** We perform supervised fine-tuning for all pre-trained LLMs (i.e., the GPT-2 and GPT-J models) on the entire training set. The prompt template used for summary generation is included in Appendix A.1.

**Preference Data, Gold and Proxy Reward Models.** To create the preference dataset, we randomly choose 100,000 prompts from the training set. For each prompt, we randomly choose two

---

[2]https://huggingface.co/datasets/CarperAI/openai_summarize_tldr

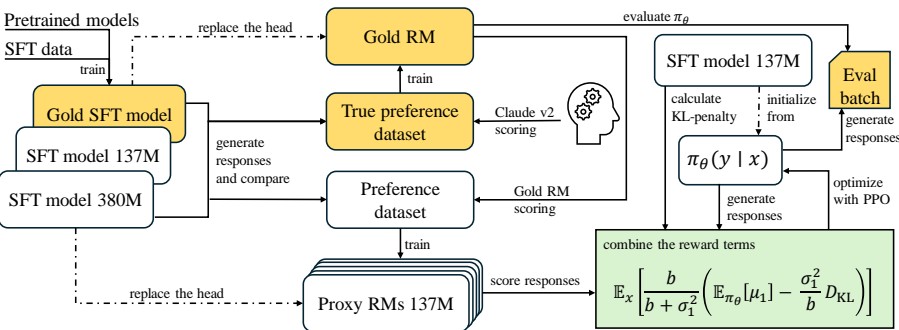

Figure 3: Our RLHF pipeline used in the experiments. We use a semi-synthetic setup, where we first train a gold reward model on true preference data and then use it to evaluate the quality of the trained LLM during the RL part.

models (from GPT-2 137M, GPT-2 380M, and GPT-J 6B) to generate summaries for comparison following Stiennon et al. (2020). Each summary within a pair is assigned a binary preference label, denoting which summary is better. We repeat this process twice using the same prompt template as the one we use for supervised fine-tuning. In the first iteration, we collect preference data to train the gold reward model (i.e., GPT-J). Each pair of generated summaries is compared by Claude v2 (Anthropic, 2023). The resulting dataset is used for fine-tuning the gold reward model. Using this semi-synthetic setup has become the de-facto standard (Gao et al., 2023; Coste et al., 2024; Zhai et al., 2023; Zhang et al., 2024b; Fisch et al., 2024; Yang et al., 2024a) as it removes large costs associated with human annotators. In the second iteration, the generated summaries are compared and assigned preference labels by the gold reward model itself. This way, we collect preference data to train our proxy reward model ensemble while having access to the "ground-truth" model, following Gao et al. (2023). In line with (Coste et al., 2024), the ensemble consists of five instances of a supervised fine-tuned GPT-2 model (i.e., either 137M or 380M). They differ in the initialization of their output layer (i.e., binary preference classification) and the batch order used in training.

**Baselines.**  We evaluate our uncertainty-aware adaptive RLHF against three baselines that follow a different definition of $\phi(x, y)$ in Eq. (2). All baselines use fixed regularization coefficient $\lambda$. The first baseline is *Standard RLHF* (Stiennon et al., 2020), using a single reward model $\phi(x, y) = \phi_1(x, y)$. The second, *Ensemble RLHF (mean)* (Eisenstein et al., 2023), uses a mean ensemble score of five reward models to define $\phi(x, y) = \frac{1}{M} \sum_m^M \phi_i(x, y)$. This is the main point of comparison, as the only difference is our adaptive $\lambda$ coefficient. Finally, *Ensemble RLHF (pessimistic)* defines $\phi(x, y) = \min_{m \in [M]} \phi_m(x, y)$ as the minimal reward out of all reward models, corresponding to the *worst-case optimization* method in the work by Coste et al. (2024).

**Implementation Details.**  We train all models by applying Low-Rank Adaptation - LoRA (Hu et al., 2021) to all linear layers with rank $r = 8$, dropout of 0.1, and $\alpha = 32$. For supervised fine-tuning, we use a learning rate of $7 \times 10^{-5}$, Adam optimizer (Kingma & Ba, 2017), a cosine scheduler with 50 warmup steps, batch size of 128. We train the models over one epoch with mixed precision on the entire training set. In all stages, we generate summaries by using top-$p$ sampling, with $p = 1$ and temperature set at 1, sampling between five and 48 tokens. For all reward models, we use a learning rate of $3 \times 10^{-5}$, Adam optimizer, cosine scheduler with 20 warmup steps, batch size of 64. We train the models over one epoch. One training step consists of sampling 512 rollouts of prompt-response pairs from $\pi_\theta$, scoring the pairs according to $\phi(x, y)$, and then applying the PPO algorithm (Schulman et al., 2017) to $\pi_\theta$. PPO goes over each batch of rollouts four times in mini-batches of six. We use the Adam optimizer and set the weight decay at 0.1, the initial KL coefficient at $\lambda = 0.005$, the clipping range at 0.2, and the learning rate at $5 \times 10^{-6}$. We tried out multiple KL coefficients $\lambda \in \{0.005, 0.01, 0.05\}$, and our findings hold for each $\lambda$ value. We run RL optimization for 50,000 steps.

**Evaluation.**  The quality of the LLM response $y$ for the given prompt $x$ is evaluated using a large GPT-J 6B reward model $\phi(x, y)$ in line with prior work (Gao et al., 2023; Coste et al., 2024; Zhai

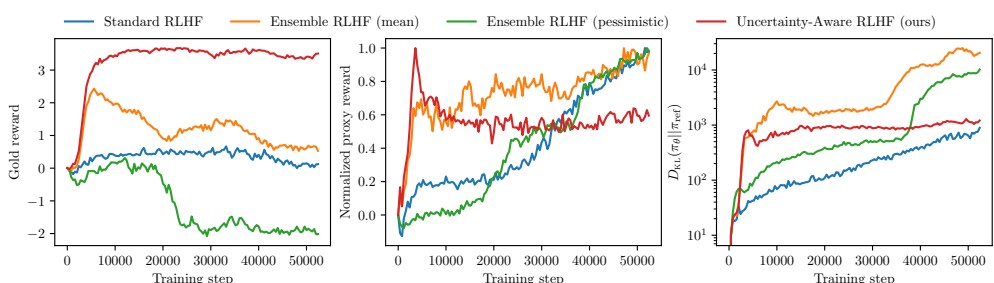

Figure 4: Comparison of our uncertainty-aware adaptive RLHF against baselines. Our method improves more, faster and does not regress during optimization (left plot). It does not increase its proxy reward indefinitely and can be used for early stopping (middle plot), and implicitly keeps the KL-divergence at a manageable distance (right plot).

et al., 2023; Zhang et al., 2024b; Fisch et al., 2024; Yang et al., 2024a). Every 400 steps of RL updates, we evaluate $\pi_\theta$ on the held-out evaluation set with the gold reward model, averaging the reward across the 2,000 prompts in the held-out data.

## 5  RESULTS

Figure 4 shows the results for optimizing GPT-2 (137M). The left plot shows how the gold reward evolves during training, the middle plot shows the proxy reward optimized by the models, and the right plot shows how KL-divergence evolves.

**Uncertainty-aware RLHF yields 50% improvement over standard RLHF.**  Our uncertainty-aware objective (Eq. 7) gives large weights to the prompt-response pairs $(x, y)$ when the variance of $\phi(x, y)$ is low. In other words, our method makes larger steps when we are sure about the reward and smaller steps when the reward uncertainty is high. This essentially reduces noise in the rewards. Because of that, we observe in the left plot, Figure 4, the LLM learns faster and squeezes more improvement from the reward model overall.

**Early stopping.**  The left plot in Figure 4 shows our method does not degrade across iterations, unlike others, making early stopping feasible. Moreover, in the middle plot, we see our proxy reward does not increase indefinitely, unlike other methods. Once $\pi_\theta$ gets too far from $\pi_{\text{ref}}$, the uncertainty-aware regularization coefficient increases substantially, training converges, and we can stop optimization.

**Pessimistic RLHF is not consistent.**  We find that the *Ensemble RLHF (pessimistic)* underperforms. First, pessimism fails when the reward model disagreement rate is too high. Eisenstein et al. (2023) argue the main benefit of ensembles is due to their reduced variance on the mean reward. Pessimistic optimization completely discards this information and cherry-picks the most pessimistic model, which is arguably the one with the highest variance. For example, if $\pi_\theta$ generates a very good response that gets high scores from four reward models but one very bad score from the last reward model, the pessimistic optimization will discourage this response in the future. This could be addressed by instead estimating the reward's lower confidence bound, but we would need to tune the confidence interval width. Our theoretically grounded method shows how to use uncertainty in a principled way, without hyper-parameter tuning, and has proven to work well in similar conditions that RLHF has (Hinton, 1999).

**Reward model ensembles are computationally tractable.**  Optimizing $\pi_\theta$ is computationally intensive. However, the additional time required to score the response by multiple reward models is relatively inexpensive. We ran our experiments using 4x NVIDIA A10 Tensor Core GPUs. Our Uncertainty-Aware RLHF with five proxy reward models takes $\sim$60 hours to complete, whereas the standard RLHF takes $\sim$56 hours, only a 7% increase in computation costs. The reward model training is also relatively inexpensive, $\sim$30 minutes to train a single reward model. Although we

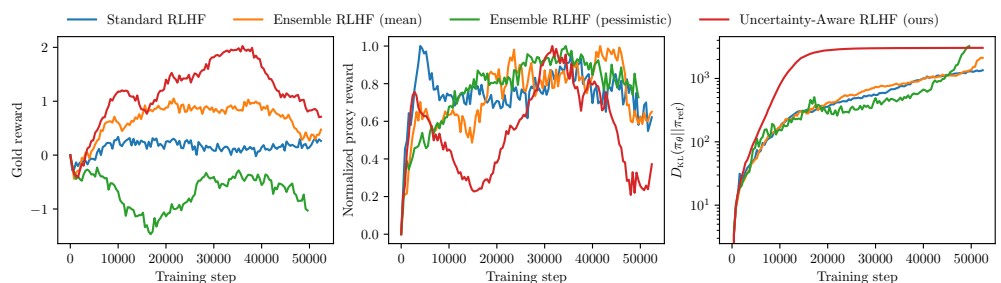

Figure 5: Comparison of our method against baselines by using a poorly calibrated reward model ensemble (GPT-2 380M).

use ensembles to measure uncertainty, our method can make use of other less expensive uncertainty estimation techniques instead (Zhai et al., 2023; Zhang et al., 2024b).

**Uncertainty-aware RLHF is robust to miscalibrated reward models.** We also experimented with a larger GPT-2 380M, where the training data is not enough to get well-calibrated uncertainty estimates. Hence, as $\pi_\theta$ moves out of the initial distribution, all reward models exhibit a systematic error in the same direction, and the ensemble variance does not increase. Figure 6 shows that while mean reward variance correlates with $D_{KL}$ in the case of GPT-2 137M, it stays roughly the same with the bigger model. However, our method still outperforms other RLHF baselines as shown in Figure 5. Even with biased reward models, our method exploits the little amount of available information and provides 100% improvements over standard RLHF. One way to improve the uncertainty calibration is to also perform fine-tuning, or even pre-training, for each reward model in the ensemble. This is out of the scope of our work since it is costly and has already been explored (Eisenstein et al., 2023).

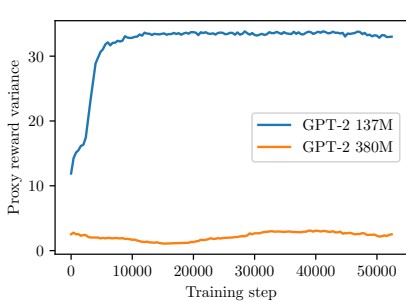

Figure 6: Variance of the ensemble proxy reward scores with GPT-2 137M and 380M.

**Qualitative Examples.** Table 1 shows an example source Reddit post and summaries generated by models optimized using the baselines and our uncertainty-aware RLHF approach. We observe that all baseline models are susceptible to reward hacking, generating repeated tokens that score highly with proxy reward models. For example, *Standard RLHF* repeatedly starts every summary with a hallucinated introduction that includes a random U.S. state, *Ensemble RLHF (mean)* repeats the word "TITLE" from the Reddit post template, and *Ensemble RLHF (pessimistic)* repeats the word "jazz" in all of its responses. See Appendix C for more examples.

## 6 RELATED WORK

**Reinforcement learning from human feedback.** Early work of RLHF focused on its application to continuous control domains (Christiano et al., 2017). Since then, its focus has shifted to aligning LLMs (Ziegler et al., 2020) on a particular task such as summarization, question answering, and web crawling agents (Stiennon et al., 2020; Ouyang et al., 2022; Nakano et al., 2022). RLHF is also used to align generic AI assistants across a variety of tasks (Touvron et al., 2023). The most popular reinforcement learning optimizer is PPO (Schulman et al., 2017), although also other algorithms have been in use, such as Implicit Language Q Learning (Snell et al., 2023) and REINFORCE (Williams, 1992). Recently, Direct Preference Optimization (DPO) techniques that use preference data without training an explicit reward model emerged (Rafailov et al., 2023; Ivison et al., 2023). However, multiple works identified issues with DPO, for example, weak regularization (Azar et al., 2024), reduction of LLM's likelihood to generate the chosen response (Pal et al., 2024), and shifting

| Reddit Post | Generated Summaries |
|---|---|
| SUBREDDIT: r/running TITLE: One year post-pregnancy, aches and pains in hips and knees, can't even comfortably do day 1 of C25k. Help? POST:[...] I've always been an on again/off again (very casual!) jogger, typically doing 3 - 5 k My knees have always been finicky, and I went to a physio who thought I had "runner's knee" [...] I had my baby a year ago, so all in all I haven't run for about 1.5 years. [...] Has anyone had something similar? Can anyone recommend some stretches or exercises that might help? Should I see a Dr? Or should I just see a physio? | **Standard RLHF** I'm an African American male, born in Florida. [...] to college in Southeast Florida, [...] |
| | **Ensemble RLHF (mean)** TITLE: One year post-pregnancy TITLE: One year post-pregnancy TITLE: One year post-p pregnancy [...] |
| | **Ensemble RLHF (pessimistic)** I'm a young lady in high school (20 - 30 years old) who loves to be involved in jazz jazz jazz jazz jazz jazz jazz [...] |
| | **Uncertainty-Aware RLHF (ours)** I've always been an on again/off again (very casual!) jogger, typically doing 3 - 5 k. My knees have always been finicky, and I went to a physio [...] |

Table 1: Example of generated summaries at the end of RL optimization.

the probability mass to responses that never even appeared in the training set (Fisch et al., 2024). On the other hand, explicit reward modeling, which is the core part of RLHF, can be easily regularized and allows better control over the alignment procedure. A direct comparison of our method with DPO is outside the scope of our paper.

**Reward hacking in RLHF.** Avoiding reward hacking when using a proxy reward model is a central problem to RLHF (Lambert & Calandra, 2023). Gao et al. (2023) studied the scaling laws behind reward model overoptimization, showing that a larger reward model and dataset size help to delay reward hacking. Naturally, this leads to measuring the reward model uncertainty as a better technique of regularization than KL-divergence. Multiple concurrent studies (Coste et al., 2024; Zhai et al., 2023) have shown modeling uncertainty with ensembles can help mitigate reward hacking by using a pessimistic optimization approach (Buckman et al., 2020). Others model uncertainty using the final embedding layer (Zhang et al., 2024b), a Bayesian reward model (Yang et al., 2024a), semantically contrastive text prompts (Kim et al., 2023), or using Monte Carlo dropout (Wang et al., 2024a). Some works argue the main benefit of ensembles is their better mean estimation (Eisenstein et al., 2023) and the difference between optimizing mean and lower confidence bounds of such ensembles is negligible (Zhang et al., 2024a). Related work has also investigated how to improve the robustness of DPO with uncertainty estimates (Fisch et al., 2024; Liu et al., 2024; Huang et al., 2024). Concurrently, Zhou et al. (2024) adds prior constraints to the reward model training, such as length ratio and cosine similarity between outputs of each comparison pair, which can reduce reward hacking in both RLHF and DPO. Wang et al. (2024b) motivate their alignment to multiple objectives by emphasizing improvements of poorly performing outputs rather than outputs that already scored well. Yang et al. (2024b) regularize the hidden states while training the reward model to preserve language modeling capabilities. Our ensemble baselines cover uncertainty methods proposed by Coste et al. (2024); Zhai et al. (2023), and mean reward variance reduction method (Eisenstein et al., 2023; Coste et al., 2024). Uncertainty penalty might be difficult to implement in practice as it requires setting the confidence interval width, an important hyper-parameter that is difficult to correctly identify. Our approach starts from theoretical insights of using the probabilistic interpretation of combining the reward and KL-divergence regularization terms as PoE (Hinton, 1999) while being surprisingly easy to implement in practice without any additional hyper-parameter tuning.

## 7  CONCLUSION

We introduced a method that mitigates overoptimization in RLHF by adaptively adjusting the KL-divergence regularization coefficient based on reward uncertainty. We derived the solution from PoE,

limiting gradients in uncertain responses, stopping the training early before reward hacking occurs. Our RLHF objective is easy to implement in practice. Empirical results on a standard summarization task show uncertainty-aware adaptive RLHF yields additional performance improvements and mitigates overoptimization. Even when the reward model uncertainty is poorly calibrated, our method method remains robust.

## LIMITATIONS

**Languages.** Our research is currently limited to English due to computational constraints and the availability of pre-trained models and preference datasets. Expanding to other languages with different characteristics presents a potential area for future research.

**Tasks, datasets, and models variety.** The RL runs reported in Section 5 took approximately 620 hours to complete (i.e., using four NVIDIA A10) with an approximate total cost of \$3,500 on a commercial cloud provider (i.e., AWS EC2 g5.12xlarge). As RLHF experiments are computationally expensive, the scope of evaluation is limited to a single task. This is in line why the experimental setting of the majority of recent related work, evaluating RL methods on a single task, dataset, and one suite of models (Coste et al., 2024; Zhang et al., 2024a; Fisch et al., 2024; Wang et al., 2024b; Yang et al., 2024a). Note that adding an additional dataset would double the costs, exceeding our budget.

**Scaling to larger models.** In Section 5, we show that our method works best when the reward model uncertainty is well calibrated (although not strictly limited otherwise). This is arguably easier to achieve with smaller models as fine-tuning larger models requires significantly more data or pretraining the models with a different random seed (Eisenstein et al., 2023). A potential workaround is to initialize the reward models from different suites of models. However, this requires additional engineering effort, and we will leave it for future work.

**Our experimental pipeline can be simplified.** When we began our study, we chose to use Claude v2 as the gold reward model. This was later shown to be too expensive, so we switched to training our own gold reward model. We used already collected preference data by Claude v2 to train this reward model. However, this step can be omitted in this setting, and instead, we can use already available preference labels for the TL;DR dataset (Stiennon et al., 2020). However, we believe this does not affect our findings; it only decreases the amount of engineering effort and cost of the evaluation.

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

## A PROMPT TEMPLATES

### A.1 GENERATING REPONSES

The prompt template shown below is used to generate the LLM responses in all stages. The LLM is fine-tuned on the data following the same format.

```
Subreddit: {subreddit}
TITLE: {title}
POST: {post}
TL;DR:
```

### A.2 COLLECTING PREFERENCES

```
Human:  You are a helpful assistant that selects the best summary out of two answers.  The summary is
good if it is accurate, coherent, and covers the most important parts of the text.  The summaries are
presented in a random order.  Write a response with the number that corresponds to a better summary
without any additional text.  For example, <doc_1_chosen_summary>2</doc_1_chosen_summary> means that for
the first document, the second summary is better while <doc_2_chosen_summary>1</doc_2_chosen_summary>
means that for the second document, the first summary is better.

<doc_1>
Subreddit:  r/relationships
TITLE: Screwed up with boss...  what should I do?
POST: I'm 20 f, my boss is around 50 years old, also f.  So I have two jobs, and the schedules for
both jobs are made on a weekly basis.  One of my jobs I have had for three years, the other one I
have had for a month and a bit.  I forgot to give my schedule from one job to my boss at my other
job, and so I was not scheduled for this week.  I didn't realize why I had not been put on the
schedule until now.  My question is, since I royally screwed up, what can I do to redeem myself?  I
don't want to call my boss today because it is a Sunday and she has the day off.  Mistakes aren't
easily forgiven where I work, as far as I can tell, and the boss often makes comments about how the
employees should be scared of her.  I have screwed up at previous jobs (little things) but my boss
was less intimidating than my current one, so I am not sure how to handle this situation.
TL;DR:
</doc_1>
<doc_1_summary_1>
screwed up at work by not giving the boss my schedule from my other job, am not scheduled this week,
what should I say in order to apologize to my (scary/intimidating) boss?
</doc_1_summary_1>

<doc_1_summary_2>
Screwed up with boss...  what should I do?
</doc_1_summary_2>

Assistant:
<doc_1_chosen_summary>1</doc_1_chosen_summary>

Human:
<doc_2>
{prompt}
</doc_2>

<doc_2_summary_1>
{response_1}
</doc_2_summary_1>

<doc_2_summary_2>
{response_2}
</doc_2_summary_2>

Assistant:
<doc_2_chosen_summary>
```

Similar to other works (Dubois et al., 2023), we replaced the human annotators with one of the most advanced LLM assistants, namely Claude v2 (Anthropic, 2023). Our template is motivated by the one of Dubois et al. (2023), and we adjusted it from evaluating general instructions specifically to choose a better text summary. We also replaced the examples in the prompt with those found in Tables 24 and 25 of Stiennon et al. (2020), which also provide example scores from human annotators. The template below shows only one example (from Table 24), then another example (Table 25) follows it, and then two other documents with their summaries follow, and the LLM assistant is asked to annotate them. To remove the position bias, we randomly choose which response is labeled as <summary_1/> for each document.

# B PROOFS FOR SECTION 3

## B.1 DETAILED DERIVATION OF EQUATION 3

$$\mathbb{E}_{\pi_\theta(y|x)}\mathbb{E}_{p_1(r|x,y)p_2(r|x,y)}r =$$

$$\mathbb{E}_{\pi_\theta(y|x)}\int \mathcal{N}(r; \mu_1, \sigma_1^2)\mathcal{N}(r; \mu_2, \sigma_2^2)r dr =$$

$$\mathbb{E}_{\pi_\theta(y|x)}\int \mathcal{N}(r; \frac{\sigma_2^2\mu_1 + \sigma_1^2\mu_2}{\sigma_1^2 + \sigma_2^2}, \frac{1}{\frac{1}{\sigma_1^2} + \frac{1}{\sigma_2^2}})r dr \quad \text{(Product of Gaussians)} \tag{8}$$

$$= \mathbb{E}_{\pi_\theta(y|x)}\frac{\sigma_2^2\mu_1 + \sigma_1^2\mu_2}{\sigma_1^2 + \sigma_2^2} \quad \text{(Expectation under a Gaussian is the mean).}$$

## B.2 PROOF OF EQUATION 5

$$\arg\max_\theta \mathbb{E}_x\mathbb{E}_{\pi_\theta(y|x)}\frac{\sigma_2^2\mu_1 + \sigma_1^2\mu_2}{\sigma_1^2 + \sigma_2^2}$$

$$= \arg\max_\theta \mathbb{E}_x\mathbb{E}_{\pi_\theta(y|x)}\frac{b\phi(x,y) - a\log\frac{\pi_{\text{ref}}(y|x)}{\pi_\theta(y|x)}}{a+b} \quad \text{(Plugging in values from Eq. 4)}$$

$$= \arg\max_\theta \frac{b}{a+b}\mathbb{E}_x\mathbb{E}_{\pi_\theta(y|x)}[\phi(x,y) - \frac{a}{b}\log\frac{\pi_{\text{ref}}(y\mid x)}{\pi_\theta(y\mid x)}] \quad (a \text{ and } b \text{ independent of } x) \tag{9}$$

$$= \arg\max_\theta \mathbb{E}_x\mathbb{E}_{\pi_\theta(y|x)}[\phi(x,y) - \frac{a}{b}\log\frac{\pi_{\text{ref}}(y\mid x)}{\pi_\theta(y\mid x)}] \quad (a \text{ and } b \text{ independent of } \theta)$$

$$= \arg\max_\theta \mathbb{E}_x[\mathbb{E}_{\pi_\theta(y|x)}\phi(x,y) - \frac{a}{b}\mathbb{E}_{\pi_\theta(y|x)}\log\frac{\pi_\theta(y\mid x)}{\pi_{\text{ref}}(y\mid x)}]$$

$$= \arg\max_\theta \mathbb{E}_x[\mathbb{E}_{\pi_\theta(y|x)}\phi(x,y) - \frac{a}{b}D_{KL}(\pi_\theta(y\mid x)||\pi_{\text{ref}}(y\mid x))].$$

## B.3 APPROXIMATIONS COMPUTING $\sigma_1^2$

**Approximation 1:** We approximate the variance of the reward models' outputs given generations from form the model $y \sim \pi_\theta(y \mid x)$, as its expectation over all generations $y$:

$$\sigma_1^2(x,y) = \frac{1}{M}\sum_m^M (\bar{\phi}(x,y) - \phi_m(x,y))^2 \approx$$

$$\mathbb{E}_{\pi_\theta(y|x)}\frac{1}{M}\sum_m^M (\bar{\phi}(x,y) - \phi_m(x,y))^2 = \sigma_1^2(y). \tag{10}$$

This approximation assumes that the variance of the reward model for a given prompt $x$ is approximately the same for all LLM outputs $y$. Note that we do not make this assumption about the mean. This approximation results in $\sigma_1^2$ to be independent of $y$ and allows us to bring it out of the expectation in the derivation of our final objective (see Appendix B.4 below).

**Approximation 2:** We introduce the stop-gradient operator over the generative model $\pi_\theta(y \mid x)$, when computing the variance $\sigma_1^2$:

$$\mathbb{E}_{\pi_\theta(y|x)}\frac{1}{M}\sum_m^M (\bar{\phi}(x,y) - \phi_m(x,y))^2 \approx \mathbb{E}_{\pi_\theta(y|x)}\frac{1}{M}\sum_m^M (\bar{\phi}(x,y) - \phi_m(x,y))^2. \tag{11}$$

This approximation results in the gradient updates not to propagate to the generative model $\pi_\theta(y \mid x)$ through the variance of the reward. This means that, during a gradient update, the variance of the reward model is first computed at the current state of $\pi_\theta(y \mid x)$ and then used in the objective function as a fixed number to perform the update. This approximation allows us to modify standard RLHF gradient updates just through re-scaling, and we can hence exploit any existing package to perform RLHF/PPO to apply our method and simply rescale adaptively the KL term.

### B.4 PROOF OF EQUATION 7

$$\arg\max_{\theta} \mathbb{E}_x \mathbb{E}_{\pi_\theta(y|x)} \frac{\sigma_2^2 \mu_1 + \sigma_1^2 \mu_2}{\sigma_1^2 + \sigma_2^2}$$

$$= \arg\max_{\theta} \mathbb{E}_x \mathbb{E}_{\pi_\theta(y|x)} \frac{b\bar{\phi}(x,y) - \sigma_1^2 \log \frac{\pi_{\text{ref}}(y|x)}{\pi_\theta(y|x)}}{\sigma_1^2 + b} \quad \text{(Plugging in values from Eq. 6)}$$

$$= \arg\max_{\theta} \mathbb{E}_x \frac{b}{\sigma_1^2 + b} \mathbb{E}_{\pi_\theta(y|x)} [\bar{\phi}(x,y) - \frac{\sigma_1^2}{b} \log \frac{\pi_{\text{ref}}(y \mid x)}{\pi_\theta(y \mid x)}] \qquad (12)$$

$$= \arg\max_{\theta} \mathbb{E}_x \frac{b}{\sigma_1^2 + b} [\mathbb{E}_{\pi_\theta(y|x)} \bar{\phi}(x,y) - \frac{\sigma_1^2}{b} \mathbb{E}_{\pi_\theta(y|x)} \log \frac{\pi_\theta(y \mid x)}{\pi_{\text{ref}}(y \mid x)}]$$

$$= \arg\max_{\theta} \mathbb{E}_x \frac{b}{\sigma_1^2 + b} [\mathbb{E}_{\pi_\theta(y|x)} \bar{\phi}(x,y) - \frac{\sigma_1^2}{b} D_{KL}(\pi_\theta(y \mid x) || \pi_{\text{ref}}(y \mid x))].$$

## C EXAMPLES OF GENERATED SUMMARIES

We present the example summaries at the end of the GPT-2 137M training of four *randomly* chosen summaries to give a qualitative overview of our method. It is clear *Uncertainty-Aware RLHF (ours)* is the least susceptible to reward hacking.

```
SUBREDDIT: r/AskReddit
TITLE: How do you get someone out of your head?
POST: Hi,
I'm 22, and I have been with my girlfriend for 5 years now.  We
recently moved together.  We've always loved each other intensely.

Problem, I recently started to have feelings for an other person
(a friend).  This person has had a boyfriend for now 3 years, and
has absolutely no ideas.  Those feelings were so strong, it was
hard to hide them.  After 2 months of me being distant and really
sad, my girlfriend forced me to say what was bothering me.  I'm not
a good liar, and now she knows.

We decided to give us a week alone, I went to my parents.

Now, I'm completely lost.  I keep on thinking about this person,
and I hate that.  I would like for those feelings to go away, to
leave me alone.  But I can't.

What do I do?  It's been 3 months now, and I'm just desperate.
TL;DR:
```

**Standard RLHF summary:**

```
I'm 23, and my boyfriend still lives at home.

I grew up in a small town, in northern Georgia, and lived in it for
few years.  Most of that time, my parents stayed in South Dakota,
```

**Ensemble RLHF (mean) summary:**
```
POST: Hi,I'm 22, and I have been with my girlfriend for 5
years now.  We recently moved together.  We've already lived
together.Problem, I recently started to have feelings for an other
person (a friend).
```

**Ensemble RLHF (pessimistic) summary:**

```
I'm scared to never go back to school, quitting school, getting
high drunk because of hookup cafes, hooking a friend, paying bills
with the few friends I have, getting known, hanging out all over
the country,
```

**Uncertainty-Aware RLHF (ours) summary:**

```
Problem, I recently started to have feelings for an other person
(a friend).  This person has had a boyfriend for now 3 years, and
has absolutely no ideas.  Those feelings were so strong, it was
hard to hide them.
```

SUBREDDIT: r/pettyrevenge
TITLE: So, my mom woke me up with a loud TV.
POST: She was in her living room, watching TV. This was at about
8:30 in the morning, and she was exercising.  She turned the TV up
extra loud to hear it over her excercycle, and woke me up.  I went
in there asking for her to turn it down.  She said she didn't have
to; I explained that I always used headphones so she didn't have
to deal with my noise and that she should give me a little more
respect, given that I paid rent at the time.

She disagreed.  I went back to my room, rather pissed off at the
lack of equality.  I had no lock on my door; but I had a dresser
right next to it, so I pulled one of the drawers out enough so that
it caused the door to not be openable.  Then, I turned my speakers
up really loud and blasted Gangnam Style on repeat, with the bass
cranked up as high as it could go.

If you hate Gangnam Style for being overplayed, you will see why
I chose that particular song.  I personally don't mind it.  But
here's the thing about my bass; it vibrates the walls, making one
hell of a lot of noise.  Needless to say, my mom was not pleased
and shut off the internet.  But it was oh so worth it.
TL;DR:

**Standard RLHF summary:**

I grew up in North Carolina, and since then I have grown up around
my sisters.  They were from California, back in the 70s.  They grew
up going to college, with my dad also living in Texas.  My father

**Ensemble RLHF (mean) summary:**

TITLE: So, my mom woke me up with a loud TV Resolution.  This was
at about 8 seconds in the morning, and she was exercising.  She
turned the TV up overtime to hear it over her exer.  I

**Ensemble RLHF (pessimistic) summary:** I took sporadic jazz jazz jazz
jazz jamming classes, fading jazz jazz jazz, skype jazz jazz jazz
jazz jamming jazz jazz jazz jazz jazz jazz jazz jamming jazz jazz
style jazz jazz jazz jazz jazz jazz jazz jamming jazz

**Uncertainty-Aware RLHF (ours) summary:**
TITLE: So, my mom woke me up with a loud TV. This was at about 8:30
in the morning, and she was exercising.  She turned the TV up extra
loud to hear it over her excercycle,

```
SUBREDDIT: r/relationships
TITLE: My girlfriend (20f) of two years cheated on me (20m) by
kissing two guys at a Halloween party.
POST: Lately her and I have been having a few problems, and these
problems have been brought up before a few times.  One problem
being that I don't show enough affection.  I don't tell her she's
pretty very often or don't compliment her much.  I feel terrible
about it, but this time I was really trying to change for her.

For Halloween she went to visit her step brother at a college and I
got drunk with my friends and watched movies.  Last night (11/1) we
got in a huge fight about me not changing and how our relationship
won't work out and basically broke up over the phone.  So in an
effort to try and fix it I drove to her house.  She told me how
at the parties she went to that two guys kissed her.  The first
one she pushed away, but the second one I asked her if she kissed
him back and she said yes and that she did it because it made her
feel wanted, which I guess I haven't been making her feel that
way lately.  We cried, we talked about everything, we had great
sex, and I stayed over at her house just to sleep with her and then
snuck out in the morning so her parents wouldn't know.

We both obviously want to work things out but aren't sure if we
should.  I love this girl, but the more I think about it, all I can
think about is her cheating on me, and more importantly, liking it.
It makes me sick to my stomach.  Should I even try to fix it or
would I be better off cutting all ties.
TL;DR:
```

**Standard RLHF summary:**
```
TL;DR: Im 20 years old.  my boyfriend lives in Colorado.  My mom
and brother live in Texas, but my grandparents live in Texas, and
me lived in Texas, and my grandparents live in Illinois, but lived
in Texas
```

**Ensemble RLHF (mean) summary:**
```
TITLE: Me
TITLE: Me
TITLE: Me
TITLE: Me
TTITLETTTT
TTITLETTTTTTTTTTTTTTTTTTITTTTITTITTIT
```

**Ensemble RLHF (pessimistic) summary:**
```
There are few things I love more than my girlfriend.

Right now I breathe weed all day, I exercise too much about jazz
jazz hip hop jazz jazz jazz jazz jazz jazz jazz fusion jazz jazz
jazz jazz jazz fusion jazz
```

**Uncertainty-Aware RLHF (ours) summary:**
```
TITLE: My girlfriend (20f) of two years cheated on me (20m) by
kissing two guys at a Halloween party.

POST: Lately her and I have been having a few problems, and these
problems
```

```
SUBREDDIT: r/running
TITLE: One year post-pregnancy, aches and pains in hips and knees,
can't even comfortably do day 1 of C25k.  Help?
POST: I'll try to keep this short!

**Background**

* I've always been an on again/off again (very casual!)  jogger,
typically doing 3 – 5 k
* My knees have always been finicky, and I went to a physio who
thought I had "runner's knee"
* Pre-pregnancy, my "runner's knee" would flare up when I got to
the 8 – 10 k distance range, even if I had a decent base (doing a
C210k type program)

**Current Problem**

I had my baby a year ago, so all in all I haven't run for about
1.5 years.  I'm quite slim and have been doing aerobics-style
classes for the past year, so I'm not totally out of shape.
Body weight exercises, aerobics, bikes and ellipticals are all
fine.  However, when I run even the tinyiest bit, or even go on
a long walk or a hike, my pelvis gets very sore and tight, and
my knees start hurting very quickly.  I already am doing general
squats/lunges/stretching type things.

I'm starting to feel like running just isn't for me anymore.  Which
is a bummer, because I really enjoy running!

Has anyone had something similar?  Can anyone recommend some
stretches or exercises that might help?  Should I see a Dr?  Or
should I just see a physio?  Not quite sure how to proceed.

Thanks!
TL;DR:
```

**Standard RLHF summary:**

```
I'm a thick African American male, born in Florida.  My grandfather
moved back to Jacksonville from Africa because he was born and
raised there.  My grandfather originally went to college in
Southeast Florida, and now lives there and lives near Cl
```

**Ensemble RLHF (mean) summary:**
```
TITLE: One year post-pregnancy
TITLE: One year post-pregnancy
TITLE: One year post-p pregnancy
TITLE: One year post-p pregnancy

TITLETITLETTTITLET
```

**Ensemble RLHF (pessimistic) summary:**

```
I'm a young lady in high school (20 – 30 years old) who loves to
be involved in jazz jazz jazz jazz jazz jazz jazz jazz, jazz jazz
jazz jazz jazz jazz jazz jazz jazz jazz jazz transition jazz jazz
jazz jazz
```

**Uncertainty-Aware RLHF (ours) summary:**

```
* I've always been an on again/off again (very casual!)  jogger,
typically doing 3 – 5 k* My knees have always been finicky, and I
went to a physio who thought I had "runner
```

