# OpenReview forum: "Adaptive Uncertainty-Aware Reinforcement Learning from Human Feedback"
_ICLR.cc/2025/Conference — Submitted to ICLR 2025_

### Official Review · Reviewer_WfcD · 2024-10-29

**Soundness:** 3
**Presentation:** 3
**Contribution:** 3
**Rating:** 5
**Confidence:** 4

**Summary:**

This paper introduces a novel approach to mitigate reward hacking in reinforcement learning from human feedback (RLHF) for large language models (LLMs). Traditional RLHF often encounters issues with overoptimization, where the model learns to exploit reward models, resulting in outputs that score well on proxies but misalign with actual human preferences. The authors propose an adaptive method that adjusts the regularization term based on the reward model's uncertainty, effectively modulating training based on confidence. They demonstrate that this method not only reduces overoptimization but also improves performance on a summarization task by 50% compared to standard RLHF.

**Strengths:**

* Originality: The uncertainty-aware mechanism is a novel solution to a well-known RLHF problem, showing how dynamically adjusting regularization based on confidence can prevent overoptimization.

* Quality: The paper’s experiments provide robust support for the method’s effectiveness, with comprehensive tests and relevant comparisons to standard baselines.

* Significance: Mitigating reward hacking while improving alignment is crucial for RLHF, especially for LLM applications where maintaining human alignment over time is essential.

* Clarity: The approach’s mathematical foundation is well-explained, and the provided algorithmic steps (e.g., Algorithm 1) are easy to follow.

**Weaknesses:**

* Computational cost: The adaptive approach involves running ensembles, which adds computational overhead. While the paper notes this is only a slight increase, exploring less expensive uncertainty estimation techniques could improve practicality. In online training, wondering how authors can load so many reward models into memories?

* Bias in the reward models (RMs): the variance estimation of reward model can be a good proxy for variance estimation. However, the error of reward model should be quantified as $\mathbb{E}_{\hat{r} \text{ trained on } D}(\hat{r}(x,y)-r(x,y))^2 = bias^2 + variance$. The paper only address the variance term and does not address the bias. Assume for example, the RM prefer longer responses with 90%, we should still debias the length instead of only tackling the variance.

* Data/prompt shift issues unaddressed: if the RM training and inference time face distribution shift in prompt, response, how does this work address that?

**Questions:**

See weakness

---

### Official Review · Reviewer_NjgC · 2024-11-03

**Soundness:** 2
**Presentation:** 2
**Contribution:** 2
**Rating:** 3
**Confidence:** 4

**Summary:**

This paper addresses the challenge of reward hacking of LLMs in RLHF. Reward hacking occurs when models over-optimize based on inaccurate signals from a proxy reward model, leading to a misalignment with true human preferences. The authors propose an adaptive uncertainty-aware RLHF approach, which incorporates reward model uncertainty into the optimization objective.

**Strengths:**

1. The approach is reasonable in its use of uncertainty-based adaptivity in RLHF. By integrating uncertainty estimation through an ensemble reward model, the paper contributes a new method to dynamically adjust optimization.
2.The paper is well-structured and clearly presents both the theoretical basis and empirical methods.

**Weaknesses:**

1. The approach assumes that reward model uncertainty, measured as ensemble variance, provides a reliable indicator of alignment confidence. However, the generalizability of this assumption across different tasks or domains remains unclear. I think a more important problem should be considered is the evolution of the reward model. The upper bound of the reward model capability is the core issue that restricts the continuous optimization of RLHF.
2. The experiments are limited to a single summarization task using the Reddit TL;DR dataset. While the results are promising, testing on a more diverse set of tasks would better demonstrate the robustness of the proposed approach.
3.

**Questions:**

Could you test the proposed adaptive uncertainty-aware RLHF approach on a more general-purpose large model, such as LLaMA 3 or other recent LLMs?

---

### Official Review · Reviewer_Svkn · 2024-11-04

**Soundness:** 2
**Presentation:** 3
**Contribution:** 3
**Rating:** 5
**Confidence:** 4

**Summary:**

The paper addresses the common issue of "reward hacking" in reinforcement learning from human feedback (RLHF). Reward hacking occurs when language models exploit inaccuracies in reward models, producing outputs that receive high reward scores but deviate from real human preferences (i.e., overoptimization). The authors propose a straightforward adaptive RLHF objective that adjusts the regularization based on reward uncertainty. The idea is to dynamically adjust the regularization weight and learning rate based on the confidence level of the reward model. Experimental results on RL;DR dataset demonstrate that this method outperforms standard RLHF approaches, maintaining alignment with human preferences and reducing the risk of over-optimization.

**Strengths:**

- Good motivation
- Straightforward approach, very reasonable for the over-optimization issue
- Good results
- Clean presentation

**Weaknesses:**

The biggest problem of the work is limited evaluation. The evaluation is restricted to a single summarization task with limited model sizes. In my opinion, the authors need to evaluate their approach on more diverse tasks and larger models, as the proposed method is a general technique for RLHF. Moreover, the main results in Figures 5 and 7 appear to be based on a single run, largely limiting the robustness of the conclusions. Repeating the experiments with different random seeds would improve the reliability of the findings. Analysis ana ablation studies are also missing.

**Questions:**

1. What is the y-axis of Figure 1? How is the "Quality of LLM" evaluated? Is it same as gold-reward defined in Section 4?
2. In Line 398, could you explain what do you mean by "well-calibrated uncertainty estimates"?
3. What is the best value of M? To what extent does the number of reward models affect the results?

---

### Meta-Review · Area_Chair_dTUF · 2024-12-18

**Metareview:**

This paper proposes an adaptive uncertainty-aware RLHF approach to mitigate reward hacking in reinforcement learning from human feedback. The method dynamically adjusts the regularization term based on reward model uncertainty, aiming to address overoptimization. Experimental results on a single summarization task suggest the method improves performance compared to standard RLHF.

Despite addressing an important problem, the work has limited scope of evaluation (a single task and small-scale models), lack of robustness testing (single experimental runs without varying seeds), and absence of analysis or ablation studies. Concerns were raised about generalizability, computational costs, and missing considerations for reward model biases and data distribution shifts.

**Additional Comments On Reviewer Discussion:**

Authors did not address reviewers' concerns.

---

### Decision · Program_Chairs · 2025-01-22

Reject